# Adaptive Route Memory Sequences for Insect-Inspired Visual Route Navigation

**DOI:** 10.3390/biomimetics9120731

**Published:** 2024-12-01

**Authors:** Efstathios Kagioulis, James Knight, Paul Graham, Thomas Nowotny, Andrew Philippides

**Affiliations:** 1Sussex AI, School of Engineering and Informatics, University of Sussex, Brighton BN1 9QJ, UK; e.kagioulis@sussex.ac.uk (E.K.);; 2Sussex Neuroscience, School of Life Sciences, University of Sussex, Brighton BN1 9QG, UK

**Keywords:** visual navigation, bio-inspired navigation, insect navigation, teach and repeat, autonomous robotics

## Abstract

Visual navigation is a key capability for robots and animals. Inspired by the navigational prowess of social insects, a family of insect-inspired route navigation algorithms—familiarity-based algorithms—have been developed that use stored panoramic images collected during a training route to subsequently derive directional information during route recapitulation. However, unlike the ants that inspire them, these algorithms ignore the sequence in which the training images are acquired so that all temporal information/correlation is lost. In this paper, the benefits of incorporating sequence information in familiarity-based algorithms are tested. To do this, instead of comparing a test view to all the training route images, a window of memories is used to restrict the number of comparisons that need to be made. As ants are able to visually navigate when odometric information is removed, the window position is updated via visual matching information only and not odometry. The performance of an algorithm without sequence information is compared to the performance of window methods with different fixed lengths as well as a method that adapts the window size dynamically. All algorithms were benchmarked on a simulation of an environment used for ant navigation experiments and showed that sequence information can boost performance and reduce computation. A detailed analysis of successes and failures highlights the interaction between the length of the route memory sequence and environment type and shows the benefits of an adaptive method.

## 1. Introduction

Ant species with individual foraging workers have impressive visual navigation capabilities despite limited sensory and neural resources [1,2,3] and can use learnt visual memories to guide them on long routes between their nest and a food source [4,5]. The ability to travel such routes guided solely by memorised visual scenes prompts the question of how such memories are organised in the small brains of ants [2] and how these capabilities might serve as inspiration for biomimetic navigation algorithms [6]. In this spirit, several papers have mimicked ant visual route navigation (for a summary, see [7]) to suggest a minimal model by which ants and robots can navigate routes by using views to recall actions rather than locations [8,9]. In these models, an agent learns a route from a newly discovered resource back to the start point of an exploratory trip by first using a Path Integration (PI)-mediated training route [10,11,12]. PI, or dead reckoning, is an innate navigational strategy by which animals continuously process their direction and speed to maintain an online estimate of the direct path back to the start of a journey [12]. The use of PI to guide an initial first route, upon which the agent learns a series of route views, is akin to “teach and repeat” procedures in robot navigation [13,14]. Subsequent navigation then proceeds by the agent rotating on the spot to find the orientation of the current panoramic scene that best matches the route memories. Because those route memories were stored when travelling facing forwards, adopting a heading that is familiar to them implicitly means that the agent is likely near the route and facing in a similar direction as when last near that location [15]. In this way, finding familiar views means recovering the correct direction to move in, implying the routes can be successfully recapitulated. These models are thus termed *familiarity-based* navigation models (see [16] as an archetype) as they differ from navigation algorithms that rely on using visual information to determine the location where the image was stored, so-called Visual Place Recognition (VPR) and SLAM methods [17,18,19].

Familiarity-based methods vary in the way that the route memories are used, being either used to train an artificial neural network (ANN) to learn a compact encoding of the route information [7,20,21,22] or instead storing all memories for individual comparison [16,23]. The latter models are termed *Perfect Memory (or, PM)* algorithms, as they retain all memories, and are used as the base model for this paper as they are easier to analyse. However, common to both types of models is that the learned route views are used/stored in an unstructured way so that the sequence they were encountered in is disregarded. Thus, with PM models, at each step, agents compare the current view to all stored route-views without any consideration of the order that those route views were stored.

However, behavioural experiments with ants have suggested that the use of context and sequence information may be part of their visual route memories [24,25,26,27]. In addition, VPR methods can be improved by the addition of sequence information (exemplified by [28], but see discussion for other examples), while SLAM incorporates temporal information through odometric estimates of position [17]. This paper, therefore, examines whether familiarity-based navigation algorithms could also be improved by the addition of sequence information, not only to benefit robotic applications but also to understand how this information could be used by ants. For simplicity of analysis (in comparison to analysing the dynamics of ANNs), a method based on PM that looks for matches only within a subset of the entire route memory set is used. This ‘window’ into the full route memory moves as the agent navigates the route, and thus, we call it the Sequential Memory Window (SMW) method. Aside from being applied to familiarity-based algorithms that have not previously had a sequence incorporated, this work differs from navigation algorithms that have used sequence in two ways (Table 1). First, a wide range of sequence lengths as well as an adaptive sequence length method are tested and strengths and weaknesses analysed. Second, in contrast to many VPR algorithms, odometric information is not used in the SMW algorithm. Odometry is left out for two reasons: first, ants can navigate routes independent of odometry and the algorithm should be relevant to them; second, a goal of the paper is to identify the benefits of sequence information independently of any benefits of odometry.

In this paper, the performance of window-based methods as compared to PM is investigated in closed-loop trials in a realistic and challenging simulation of the natural navigational habitat of desert ants [5]. A series of fixed window lengths as well as an adaptive window-size approach are tested and both the navigational success and computational costs of the algorithms are compared. As it is a closed-loop test, the marker of success is how many times the algorithms fail to follow the route, dubbed the Trial Fail Count (TFC). By analysing the strengths of window-based methods, the benefits of a sequence for performance and computation for both ants and robots are shown. However, dissecting the types of failure highlights weaknesses and suggests the need for alternative navigational modes in addition to route following.

## 2. Materials and Methods

### 2.1. Route Navigation Simulations

In this article, the results of testing navigation algorithms in a simulated 3D environment are reported. The simulation is representative of the environment of desert ants and is reconstructed from LIDAR scans of a field experimentation site near Seville, Spain [40] (the underlying data are available at https://insectvision.dlr.de/3d-reconstruction-tools/habitat3d, accessed on 18 August 2024). A top-down view of the environment and example views are shown in Figure 1. As is typical of desert ant habitats, objects are quite sparse and quite uniform in height leading to a fairly uniform view of distant objects (Figure 1b), unless the agent gets close to, or passes under, a grass tussock which leads to partially or fully occluded views (Figure 1c,d). Before being input to the algorithm, views are minimally pre-processed by downsizing and converting to greyscale (Figure 1e).

### 2.2. Familiarity-Based Navigation with the Perfect Memory Algorithm

In the navigation experiments, an agent first traverses a predefined route in the simulated environment. During the route traversal, the agent periodically stores views, or snapshots, approximately every 4 cm of movement along the route. The snapshots are stored as panoramic images together with a heading, i.e., the current direction of movement. The heading is defined as the direction from the current location to the location where the next snapshot is taken. Unlike other visual familiarity algorithms, the sequence of the snapshots along the training route is also stored. The agent subsequently navigates along the learned route by rotating on the spot through a set of headings and comparing the rotated current view to all stored snapshots using an image difference function (IDF). Once it finds the heading that produces the minimum difference across all snapshots and headings, it takes a step in this direction and repeats the process.

A large variety of IDFs have been used in the literature. Here the mean absolute error is used as the IDF:(1)$$\mathrm{IDF}\left(C\left(\vec{x},\theta \right),S\left(\vec{y},\phi \right)\right)=\frac{\sum_{i=1}^{M}\sum_{j=1}^{N}\left|C_{ij}\left(\vec{x},\theta \right)-S_{ij}\left(\vec{y},\phi \right)\right|}{MN}$$
where $C(\vec{x},\theta )$ is an M×N panoramic view from position $\vec{x}$ at an orientation θ with pixel greyscale value $C_{ij}$ at row *i* and column *j*, and $S(\vec{y},\phi )$ is a view stored from position $\vec{y}$ at orientation ϕ with pixel values $S_{ij}$. To derive the direction of travel, the PM algorithm finds the minimum of the IDF across all rotations of the current image and all memories from the training route and takes a step in the direction of the heading ($\theta_{t}$) that generated the minimum of the IDF for the best-matching snapshot, i.e.:(2)$$\theta_{t}=\underset{\theta \in \left[-180,180\right],S^{k}\in Tr}{arg min}(\mathrm{IDF}\left(C_{t}\left(\vec{x},\theta \right),S^{k}\right)$$
where Tr is the set of all the training images, and $S^{k}$ is the *k*’th element of these images.

### 2.3. Spatiotemporal Memory Window Algorithms, SMW and ASMW

Unlike the perfect memory algorithm, the sliding memory window (SMW) algorithm adapted from a preliminary model in [23] compares the agent’s current view to a subset, or *window*, of size *w* around an estimated current position, the *Memory Pointer* ${\mathrm{MP}}_{t}$ (Figure 2). That is, comparisons are made with training snapshots $W_{t}=\left\{S^{{\mathrm{MP}}_{t}-\frac{w}{2}-1},\ldots ,S^{{\mathrm{MP}}_{t}+\frac{w}{2}}\right\}$. In addition to determining the agent’s movement direction, the best-matching view also determines the position of the memory window in the snapshot sequence for the next navigation decision:(3)$$\theta_{t},{\mathrm{MP}}_{t+1}=\underset{\theta \in \left[-180,180\right],S^{k}\in W_{t}}{arg min}(\mathrm{IDF}\left(C_{t}\left(\vec{x},\theta \right),S^{k},{\mathrm{MP}}_{t}\right),$$
where $C_{t}$ is the query view and $\theta_{t}$ and ${\mathrm{MP}}_{t}$ are the optimal rotation angle of the query view and the centre of the memory window for the next comparison, respectively. The length of the window, $w=|W_{t}|$, is a free parameter, and a range of fixed lengths were tested, namely: 10, 15, 20, 25, 30, 40, 50, 75, 100, 150, 200, 300 as well as two methods where the length of the window adapts. Each algorithm and window length is tested on 20 routes of increasing curvature with the performance averaged across three trials.

In the adaptive sliding memory window (ASMW) algorithm, the window starts at a particular size (here 15 and 20 are tested), but the size of the memory window is adjusted as a function of the change $\Delta_{t}=m_{t}-m_{t-1}$ of the minimum of the RIDF. That is, if the difference between images is decreasing, the window reduces and vice versa. Specifically,
(4)$$m_{t}=\min_{\theta \in \left[-180,180\right],S^{k}\in W_{t}}(\mathrm{IDF}\left(C_{t}\left(\vec{x},\theta \right),S^{k},{\mathrm{MP}}_{t}\right).$$

The adaptive window size $a_{t}$ is then adjusted according to
(5)$$a_{t+1}=\left\{\begin{array}{cc} \min\{a_{t}+\left\lceil \frac{w_{\min}}{\ln(a_{t})}\right\rceil ,w_{\max}\}\; & \mathrm{if}\;\;\Delta_{t}\geq 0\;\\ \max\{a_{t}-\left\lceil \ln\left(a_{t}\right)\right\rceil ,w_{\min}\}\; & \mathrm{otherwise} \end{array}\right.$$
where $w_{\min}=10$ is the minimum window size for the experiments in this work and $w_{\max}=600$, which is the total number of training images.

The window is initialised at the start of the route, with size $a_{0}=15$ or $a_{0}=20$ and with the memory pointer pointing at the first index. In an alternate scenario (e.g., the kidnapped robot problem), a full memory scan could be conducted to set the initial window position, meaning $MP_{0}$ could be anywhere within the training route. However, as the agent is known to begin at the start of the route, we decided against this.

### 2.4. Agent Movement and Trial Fail Count, TFC

At the start of each trial, the agent is placed at $P_{start}∼N\left(\mu ,\;\sigma \right)$, where $\mu =P_{route,1}$ is the start of the training route and σ=10 cm. The agent then derives a direction from one of the navigation algorithms and moves a set distance of 5 cm in that direction. Then, a new view is rendered and the process repeats. Note that the agent does not have collision avoidance (as ants would move between the tussocks that make up the objects in the environment); hence, the movement is not affected by obstacles.

Following Ardin et al, [20], to assess the performance of agents that are navigating autonomously, the Trial Fail Count (TFC) metric is used to count the number of times an agent fails on a given route. The TFC is increased under either of the following two conditions. First, if the agent diverges from the route by more than a threshold $d_{\max}=$ 30 cm, the trial is stopped and TFC is incremented by 1. To determine the distance from the route, the minimal Euclidean distance between the agent and training route snapshot locations is used. The TFC is also incremented when the agent does not progress sufficiently along the route. Every 10 timesteps of the simulation, the index of the training route position that is nearest to the agent’s current location, $V_{t}$, is checked. The search for the nearest route point always starts from the checkpoint ten timesteps ago $V_{t-10}$ and the nearest route point becomes the current checkpoint and measures the progress of the agent along the route in training route images. If the agent has not progressed sufficiently in 10 time-steps, that is, that $V_{t}\leq V_{t-10}$, TFC is incremented by 1. This second rule ensures that the agent does not remain ‘stuck’ in one region in the vicinity of the route but below the distance threshold.

If either of the failure conditions occurs, the agent is repositioned to the location corresponding to 5 index places (approximately 20 cm) in advance of the $V_{t}$. This ensures that the agent will not repeat the exact same failure again. The window memory pointer $MP_{t}$ is also reset to $V_{t}$. Algorithm performance is assessed as the mean TFC across three trials over 20 routes of increasing curvature, giving 20 mean TFC values. Each of the three trials starts at a different random starting position for each route. However, each algorithm variant is given the same starting position for each trial-route combination. Three trials were deemed to be sufficient as the reset behaviour following failure means that three results are representative of general behaviour.

## 3. Results

Our goal is to evaluate window-based methods of visual familiarity navigation in a realistic simulation of a desert ant’s habitat, with simulated agents controlling their path in a closed loop. To do this, the agent takes a step in the direction determined by each algorithm and repeats unless it either finishes the route, gets stuck (i.e., starts to oscillate around a point) or veers too far from the route. If either of these failure modes occurs, the agent is placed back on the route near the point it diverged, and the *Trial Fail Count*, or TFC, is incremented (see Methods for details). The performance of all methods is first compared, before the strengths and weaknesses of window-based methods are assessed, with the paper ending with an analysis of ASMW performance.

### 3.1. Closed-Loop Performance of Sequence-Based Algorithms

Results comparing the PM algorithm, which has access to all route memories at every step, to window-based algorithms with windows of differing lengths (henceforth, referred to as SMW(20) or SMW(100) where the (number) is the window length) are presented first. In addition, the performance of two adaptive window methods, ASMW(15) and ASMW(20), in which the length of the memory window grows and shrinks depending on the current best match, are assessed. In the adaptive methods, “15” and “20” denote the starting window length. The algorithm variants are tested on 20 routes of increasing complexity and curvature. For each route, performance is calculated as the mean TFC across three start positions (which are the same for each algorithm). As there was no systematic variation in accuracy for different algorithms with route shape (see below), the distribution of mean TFC across all 20 routes is shown in Figure 3a. Overall, the majority of moderate window sizes do well and the worst-performing methods are the PM algorithm (PM is significantly worse than SMW(20), p<0.001 and ASMW(15), p<0.01, paired sign-rank test) and the window-based methods with the shortest (10) and longest (300 and 500) window lengths. For moderate sizes, there is not too much sensitivity to the exact size, with the best performance being for window sizes 20, 25 or 50 if one considers the median, 25th or 75th percentiles, respectively. The performances of the adaptive windows are good but not optimal and in the middle of the better-performing windows (no significant difference between medians of ASMW(15) and SMW(20)). This is to be expected: there is likely always a window size suited specifically to a given environment or part of an environment. There is little difference between the two adaptive variants, indicating that the starting window size is not critical.

As illustrated in Figure 3b, as well as performing better, the window-based methods also require significantly less computation than PM. While PM compares every test image with all route images, the windowed methods only compare each test image to the images within the window, i.e., PM has a time complexity of O(R×θ), where *R* is the total number of route images and θ the number of in-silico rotations performed, whereas when using a window-based method, time complexity is O(w×θ), where *w* is the window length and, typically, w≪R. While the adaptive window length varies and is not capped at a maximum size, the window length generally ends up being between 20 and 100, leading to low runtimes (Figure 3b).

Intuition might suggest that routes with higher curvature would suit shorter windows and those with lower curvature, longer windows and, indeed, this was observed in preliminary work [23]. However, robustly testing this in a complex environment in a closed-loop does not reveal an obvious relationship between curvature and window size. Examining Figure 4a, in which the TFC on routes of increasing curvature is shown, there are no systematic differences in performance versus window size. While this could be because the spline-base method of route generation means that curvature is unevenly spread within the routes (see Figure 4c for examples), routes 0–3 are largely straight so should favour the longer windows but do not (and, if anything, favour shorter windows). Some routes show a more consistent pattern, but it is route-centric and is not obviously related to curvature. Specifically, routes 3 and 18 (green boxes in Figure 4) seem to have an optimal intermediate window size, while routes 6 and 11 favour long windows (red boxes in Figure 4) and route 8 seems to be best for the shortest window (purple boxes in Figure 4).

The counter-intuitive reason that dependence on curvature is less prominent than expected is owed to the interaction of the failure mode and the closed-loop ‘live’ trials. Specifically, the vegetation (tussocks) in the simulated environment (Figure 1b–e) has a significant spatial extent and idiosyncratic shapes. Failures are, in the vast majority, caused by proximity to a tussock causing occlusions and, therefore, a bad match, rather than ‘pure’ aliasing where two parts of the unobstructed world look very similar. This interacts with the live-ness of the trial because when the training or test trajectory passes through a tussock, there is a set of occluded views which (in contrast to static tests in [23]) lasts for an extended period. There is still an interaction between failure and curvature, as loops/points of high curvature increase the likelihood of aliasing, but this essentially amplifies the problems of occlusions. Hence, failures occur at similar ‘difficult’ locations in the environment as shown by similarities in the failure points for different algorithms: compare the positions of the failure points (red circles in Figure 4b) for PM (top-row), SMW(20) (middle) and ASMW(15) (bottom). While some common points are at regions of high curvature (e.g., Routes 18 and 8), others are in straight regions and driven by tussocks.

Overall the results demonstrate three things. First, algorithms using a memory window can outperform the PM algorithm despite not having access to all memories if the window size suits the route. Second, the optimum window size varies for different routes, but the worst performing overall are very long or very short. Third, an adaptive window size can ameliorate the worst results of being too long or short and performs close to the best overall suggesting that this could be a useful alternative to a fixed window method.

### 3.2. The Benefits of Sequence Information: How Can Window-Based Methods Outperform PM

Given that the PM algorithm has access to all route memories, rather than just a subset in the window methods, the question arises as to how the latter can outperform PM? Ultimately, this is because the window methods retain information on the temporal structure of the training data and, by only searching around the current estimate of position, apply (mostly accurate) knowledge to restrict the search to relevant training images. The way this helps can be seen in Figure 5a,b, where the route contains a loop. In this instance, when at the crossing point of the route, the agent experiences the same visual scene but must go in different directions depending on when in the route it experiences the scene. PM will often fail in this situation as it does not have this contextual information, and hence, in the example in Figure 5a, does not enter the loop but performs a ‘short-cut’. In contrast, most window-based methods do not fail because they only match a subset of memories, meaning that, for this loop, the window does not contain images from both visits to the crossing point, and hence, the route is completed correctly (Figure 5b).

Window methods can only help with loops if, however, the window is short enough that it does not cover the whole of the loop, meaning small loops can cause issues for longer windows. For instance, examining the performance of a window of length 300 on a route with a small loop shows that there are several failures close to the small-looped portion of the route (red circles in Figure 5c). The latter three of these failures are caused by aliasing due to the training route (black line) having a short looped portion. This can be seen in the heatmap (Figure 5d), which shows the minimum difference (across all headings) for every query image (on the y-axis) with every training image (x-axis) as a colour: blue is low = a good match; yellow is high = a bad match. Differences between images in the looped portion of the training and test routes (see the upper right of Figure 5c) are visible in the centre of the heat map, where two sets of good matches form a dark blue cross-like structure. The optimal match (i.e., the match the agent should take; dashed white line) follows the correct arm of the loop, whereas the algorithm matches with the wrong arm and ‘jumps’ to the wrong side of the cross at rows 220 and 270, leading to failures. While harder to see from the figure, the failure at row 250 is also caused by the loop as the agent gets stuck in a small looped movement due to the crossing point at row 250.

It could of course be argued that such small loops are unrealistic in that an agent would rarely need to perform such a high curvature movement (and it is not a behaviour typically observed in foraging ant paths). However, failures of PM also occur at non-loop route segments where many shorter windows do not fail. The reason for this can be seen by examining the position at which the agent veers off route (Figure 6a). As mentioned previously, failures typically occur when distant objects in the current query image are occluded—in this case by going too close to a small nearby tussock (yellow arrow in Figure 6a) seen in the right half of the test view (Figure 6b, top row). Because occluded parts do not give useful information, matches can occur either with another image that has the same level of occlusion—leading to aliasing—or with the subpart of the image where the same objects can be seen in training and testing.

Comparing the match from the occluded test view with a nearby point on the route reveals a low-quality match (i.e., a shallow minimum) driven by objects in the left half of the images while the occluded right side does not match at any heading, meaning the RIDF is high for all headings (Figure 6b, right). However, as the matched parts are the same objects, the derived heading is correct. Because PM can match with any memory from the training route, when the nearby match is only partial, there is a chance that it will match better with part of an image from a distant position where the occluded part matches. In this case, the shape of the leaves from a random tussock from position 190 in the training route (Figure 6b, left) makes a decent match with the occluded part of the test image. This results in a lower minimum for the RIDF than for the nearby image, despite not matching the near objects in the left half of the view and giving an erroneous heading (Figure 6b, bottom panel). Because the window constrains the matches to nearby positions, there is more chance that the best match from within the window will be driven by a partial match with correct objects rather than matching the occlusion. Thus, the window methods can outperform PM by biasing the algorithm towards the correct parts of the route and, implicitly, the correct parts of the images.

### 3.3. When Do Sequence-Based Methods Fail

Having seen the reasons why long windows and PM fail on route 16 and how short windows do well, one might think that it is desirable to use a very short window to minimise computation. However, looking at the performance of short windows overall, it is clear that making them too short is also detrimental (compare the PM and SMW(10) results in Figure 3a and Figure 7a). In contrast, for route 16, the adaptive windows perform very well with near-perfect results. To understand why this is, it is instructive to look at the failures of the shortest window. First, note that points of failure, or difficulty (as evidenced by tortuous paths) for both SMW(10) and PM are in similar locations whether the route is complex or straight (compare positions of cyan circles and yellow stars in Figure 7b,c). As mentioned, the root cause of failures is paths getting too close to tussocks, or passing into a tussock, causing either complete or partial occlusions. For complete occlusions where the views bear no resemblance to nearby views, there is no hope of a visual-only algorithm working, and the agent will generate a random heading with a random view (often aliased for longer windows), which might luckily take it out of the tussock, but more often causes random headings and eventual failure. In the case of partial occlusion, there is a chance of success if the correct part of the view is matched which, as discussed in the previous section, is more likely for shorter windows. Therefore, why do short windows fail almost as often as PM in tussocks?

The issue for short windows is evident when considering the heatmap of minimum differences between training and testing views for route 0 (Figure 8a,b). As training views are on the x-axis, a tussock encountered in training and causing bad matches will appear as a yellow vertical band (as the occluded views match badly with all rows/test views). Likewise, occluded test views generate a horizontal yellow band. As tussocks have a spatial extent, and training and test paths are similar, tussocks near the paths cause vertical and horizontal bands to coincide. However, this does not mean the agent is in the exact same position but that it is encountering the same tussock.

It can be seen that the two failures for SMW(10) are caused by tussocks, as shown in Figure 8a. The first, starting from row 235 and column 253 is particularly instructive, as it is caused by an area of partial occlusion caused by low grass, meaning there should be enough information for the agent to pass through. This can be seen in the expanded heatmap in Figure 8b where both vertical and horizontal yellow lines due to the occlusions are not solid but striated as the agent has views that are more or less occluded. In the vertical, this corresponds to the width of the tussock as the agent travels in a smooth path through it, but in the test, the agent is not moving directly forward but meandering through. Thus, until the grass, the path of the agent (red line) exactly matches the optimal position (the closest training view to that test view spatially; white dashed line). Upon entering the grass, there is a partial match with locations about one-third of the way into the tussock. This remains the best match, despite the agent moving forward, leading to a divergence between the optimal and matched positions (the white-dashed line is first behind, then ahead of the red line) so that when the agent reaches the edge of the grass (row 248) its estimated position—and crucially the centre of the window—is *behind* the current position of the agent.

At this point, the issue of a small window is revealed. The current view is not occluded, meaning it matches well with training views before and after the tussock, as shown by the minimum image difference of test image 248 with the training images near this (Figure 8c). However, as the window of potential matches surrounds the previous best match, which lags the current position, the forward edge of the window does not extend outside the occluded views while the views behind extend before the grass. At the next step, the match occurs with a view from before the grass, moving the memory pointer to this position. The window is now composed of images from before the grass and occluded views forward, meaning that from this point on, as the current view is not occluded, there will only ever be matches before the tussock, resulting in the window getting ‘stuck’ and a failure resulting due to a lack of forward progress after 10 steps (row 258).

The issue of lagging windows can also be seen for the next point of failure (Figure 8e,f). Here, there is again a minor occlusion as the path goes near a tussock (yellow arrow in Figure 8e) causing view 368 to match behind the true position and find a locally optimal best match with column 378 (Figure 8f). The window is stuck at this point as it cannot get past this local minimum in image difference space as the front edge just reaches the small tussock, causing the vertical yellow line at column 383 in the heatmap (Figure 8f). Despite the poor match, the agent moves forward, but crucially, the window does not, meaning the matches become worse, leading to divergence from the route and failure (cyan line and circle in Figure 8e).

Thus, while sequence information can improve performance, the benefits clearly rely on the information being correct. If the window surrounds the wrong portion of the route, there is no chance of matching, meaning that, as the window position is set by the current match, small errors can be catastrophic, which is more likely for shorter windows. The window can also jump ahead of where the agent is, meaning that the window again surrounds the wrong location and leads to failures. However, as the agent has a chance to ‘catch-up’, the issue is not as bad as for ‘lagging’ short windows, resulting in fewer errors (Figure 8d).

### 3.4. Does the Adaptive Window Method Improve Performance

Given that a window can be too short or too long depending on the interaction of environment and route shape, which often change during the route itself, the ASMW algorithm was introduced. In this method, the size of the window adapts during navigation, adjusting access to route information in real time depending on the route or environment. The intuition was that if the agent diverges away from the route due to bad matches, or gets stuck, the window will grow as the similarity drops and it will search more memories until the window is equivalent to a full PM search or the agent converges back to the route. Once the agent nears a familiar location near the route, the image familiarity grows and the window shrinks in size. In actuality, a failure is often triggered before the window grows very large, but a bigger window near difficult regions of the world should give the agent more chance of avoiding issues due to lagging while, by subsequently shrinking, computation and aliasing risks should be reduced by focusing on the correct set of views. As noted in Section 3.1, the ASMW performs well overall with generally promising results, but is this performance actually due to the window adapting to the demands of the environment and avoiding getting ahead of, or falling behind, the optimal match?

Because it takes exactly the same route up to and within the grass obstacle, the failures used to highlight lagging issues for a short window above (Figure 8b) illustrate the way in which the ASMW can avoid the issue of a lagging window. First note that the black and red lines are identical up to row 248 in Figure 8b as the path is the same until exiting the grass. However, the ASMW window has grown due to periods of poor matching both before and within the grass (solid black lines in Figure 8b), so that when it exits the grass, while centred at the same location as the fixed window, the front edge of the window extends to image 275, beyond the remaining occluded training views. Thus, the best match within the ASMW window is ahead of the grass as this minimum is lower than the local minimum found by SMW(10) (Figure 8c) as it is nearer the current position of the agent, meaning ASMW avoids getting stuck and progresses. In the second example of short window failure (Figure 8d,e), the adaptive again succeeds at this region partly, as it has now taken a slightly different path, but additionally, the window has grown (to 26) so would not suffer the lagging issues.

Further indications that window adaptation is beneficial are seen in a typical example of ASMW performance on route 10 (Figure 9). Here, one can clearly see that when difficult regions in the environment (Figure 9a) cause bad matches, the window grows, which can help in avoiding lagging (notice how the red window limits expand around the yellow bands, showing the bad matches in Figure 9b). Further, it can be seen that the expansion is useful by looking at the window width (i.e., half the window length; blue line in Figure 9c) alongside the difference between the indices of the training views where the algorithm should be matching (the nearest training view) and where it is matching (red line in Figure 9c). In instances where poor matches occur because the difference between the current and optimal location grows, the window generally adapts so that despite the window centre lagging or being ahead of the current match, the optimal is still within the window (the red line is below the blue line despite spiking high at times). This means that when the agent exits the bad region, the algorithm can get back on track (unless it has already strayed too far, as is the case for the final failure).

Examining the results more generally, if adapting the window size works well, the window length for ASMW might be expected to ‘track’ the ‘optimal’ window length. Given that there is no obvious optimum for many of the routes, a clear correlation is impossible, but again there is cause to think that the ASMW is working. Specifically, when ASMW does well, the mean window length is close to the optimal fixed window size, indicating that it is adapting to conditions sensibly (e.g., Routes 1, 18, 17; Figure 4). At first glance, route 14, where ASMW performs optimally, seems to buck this trend as ASMW performance is different to windows near it. However, on close inspection, near the relevant region of the route (where there are many failures for fixed windows), the window is 60, which is around the best-performing fixed windows. This highlights both adaptation and the difficulty of analysing routes that contain harder and easier regions.

The ASMW does perform relatively poorly on some routes, however. On Routes 6, 16 and 3, which favour either the longest window or a window of 100, it performs badly as do shorter fixed windows. A faster rate of window growth might overcome these issues, or starting the window at a higher initial value. Preliminary experiments with a variety of schemes for window growth were performed, with some variations in results. However, it was decided not to optimise all parameters for this particular environment so as to highlight the benefits of adaptiveness more fairly against fixed windows and PM. The other poor results are in routes 4 and 15. Both of these routes have somewhat ‘flat’ profiles due to difficult regions that ‘catch’ many of the algorithms, in the case of route 15 due to a narrow U-shaped route that gives ‘true’ aliasing. Thus, we suspect that good performance here is a matter of serendipitously overcoming one of these failure points, though it is hard to be sure. What is clearer is that at these failures, the optimal match is within the window meaning that it is not the adaptivity of the window per se that is causing issues (i.e., a fixed window would not help in these cases). However, it remains that the ASMW is not an unqualified success.

## 4. Discussion

In this paper, the computational and performance benefits of incorporating temporal information into familiarity-based navigation algorithms through route sequence information have been explored and confirmed. In particular, by comparative analysis of the successes and failures of algorithms with and without sequence information, and by focussing on the amount of temporal information available (via the window length), the ways in which sequence information can be beneficial have been dissected. Sequence information implicitly focuses the agent on the relevant portion of the world, reducing issues caused by loops in routes and meaning that in the case of partial occlusions, the agent has more chance of matching ‘good’ information. However, PM was not outperformed on every route. Indeed, the weaknesses of using sequences were highlighted by analysing how short windows can occasionally ‘lag’ behind and become hopelessly lost. While an adaptive window method can ameliorate these issues, no algorithm was completely successful, leading to the conclusion that additional navigational methods are needed. Below, the implications for navigation in robots and ants are discussed.

The improvement in performance is in line with other research demonstrating the benefits of incorporating either implicit or explicit information in sequences in navigation algorithms. For instance, several papers extended the range of single snapshot navigation by visually homing between a sequence of waypoints, which implicitly utilises sequence information [29,30,31,41]. In this case, sequence information is included in the (typically small, i.e., around 10) discrete waypoints, but was not explicitly used in the matching process; hence, a sequence is referred to as being ‘implicit’ for these methods in Table 1. While these methods can have impressive results [31], there is evidence that the use of a strict sequence of images as attractors to a location is brittle [30]. One solution to this problem is to move from homing to individual waypoints and consider methods that set the bearings for movement. Early work in this vein from Matsumoto et al. [32], Andersen et al. [33] and Tang and Yuta [34] show that navigation can be achieved with low computation by attempting to only match the next view with the current estimate plus one or two near neighbours, very neatly showing the benefits of short sequences. A third branch of work used long sequences of images to improve the robustness of visual place recognition algorithms, notably the SeqSLAM algorithm [28]. SeqSLAM matches a sequence of current views to all possible sequences in the training views with the increased complexity of matching being compensated by increased performance for lower-resolution images. Indeed, further models from the same lab ably demonstrate the utility of sequence information for VPR in a number of guises [35,36,37]. While VPR does not directly provide the guidance information robots need, teach-and-repeat route navigation algorithms have also exploited sequence information. Zhang and Kleeman [38] use a panoramic image for visual heading correction and compare it to a (quite restricted) subset of the training routes around the current best estimate of position. This paper has inspired several others, with the most recent paper showing particularly impressive performance through the integration of odometric information with sequences [39]. This final set of work is most similar to the algorithms presented in this paper but differs in the fact that the estimate of position/sequence relies on odometric information.

This paper therefore adds to the field in three ways. First, a non-odometric method of incorporating a sequence is shown to be beneficial to familiarity-based navigation algorithms. Adding odometry (e.g., to avoid big jumps in window position) would very likely help the algorithm further. However, using a non-odometric method to update the estimate of the current position was important for two reasons: (1) to keep the work relevant to ants (who do have a constantly updated odometric estimate of position but can navigate without this); (2) it allowed the benefits of sequence to be discerned separately from the benefits of odometry. The latter point was important, as the second (and perhaps more critical) contribution was to analyse how sequence information helps and, in so doing, reveal potential pitfalls. Specifically, focusing attention on the most likely relevant training data is clearly useful if the information is correct but can lead to failure if it is not. Additionally, there is a tension between how much one focuses, versus retaining a wider view. This is seen in the variation in results for different sequence lengths (Figure 4) which also indicates that the optimal amount of focus is likely specific to the interaction of route shape and the environment at that location. The final contribution was to show that a (purposefully simple) adaptive method can help in finding this balance, but this method is not perfect. On some tests, PM was the best algorithm, implying that, for those cases, navigation was not helped by having temporal information. While performance could have been improved through optimising ASMW (discussed below), it was concluded that, for the family of algorithms presented here, there needs to be an additional navigational mechanism that assesses when the information being used to navigate is wrong or uncertain. Such a confidence measure could be derived from the image-difference metric illustrated in the various heatmaps and could be used to trigger a decision to try a different path.

When considering the implications for ants, it is interesting to note that ants do have additional navigational mechanisms which seem to be triggered in moments of uncertainty and result in an increase in exploration and search for extra sensory information [42,43,44]. Thus, it seems that ants might be able to take the benefits of temporal information while guarding against some of the weaknesses. When thinking about how this might be implemented neurally, the obvious candidate would be some sort of selective attention highlighting some of the view memories (but, see [45]). The amount could be tuned by evolution to the demands of the environment, but whether the amount could be adapted, moment-to-moment, is more speculative. Whatever the neural mechanism, these results show that, at the very least, temporal information can be useful for navigating insects.

The final point of discussion is whether one should use the ASMW algorithm for familiarity-based visual route navigation. ASMW performance, while not optimal, is good and provides flexibility and robustness when the properties of the route and environment are unknown. That is, the dynamic window allows the agent to adapt to its current situation. In addition, the ASMW parameters were purposefully not optimised, as this would have biased the algorithm comparison by overfitting to the particular routes used, but this does leave room for improving ASMW (discussed briefly below). However, experiments with different window growth schemes (e.g., different weightings on the growth, making growth proportional to the gradient of image difference, using the gradient of bearing to influence growth) did not lead to large changes in performance, leading to the conclusion that, for demonstrating utility of an adaptive mechanism, the simple method was the best to use.

## 5. Conclusions

In summary, this paper has shown that incorporating sequence information into familiarity-based algorithms can improve performance and reduce computation. Further, by assessing how sequence information is used, and comparing it to what is known about how temporal information is used by insects, the insights shed light on aspects of insect neuroethology and suggest improvements for robotic navigation via further bio-inspiration. As ASMW algorithms show performance benefits in terms of computation and are suitable for robots with visual sensors only, use-cases are focussed on GPS-denied or unreliable applications, as well as those where the payload is limited such that Lidar sensors are not practical. In addition, it would be suited to environments in which recognising location accurately (such as in dynamic outdoor environments) is challenging. Use-cases therefore include: space robotics, UAVs or small robots in unstructured outdoor environments in changing weather conditions and in indoor extreme environments such as nuclear inspections or decommissioning.

As these use-cases each have their own particularities and challenges, the intention of this work was to keep the implementation simple, so that benefits of the temporal window algorithms could be thoroughly investigated and applied to both ants and robots. However, this leaves various avenues for future work, especially within robotic implementation, where more information could be incorporated depending on the robot’s sensory capabilities. For instance, optic flow-based visual processing could be used to highlight only the useful parts of the scene (typically those in the middle distance, not too near that they move quickly and not too far that they are essentially fixed). Additionally, information about self-movement could be used to more accurately update the window position in combination with the best-matched positions (similar to [39]). Furthermore, future robotic applications will incorporate obstacle avoidance (to adjust small-scale details of the path) and path optimisation (to optimise the overall trajectory over multiple traversals) into the route following procedure.

## Figures and Tables

**Figure 1 biomimetics-09-00731-f001:**
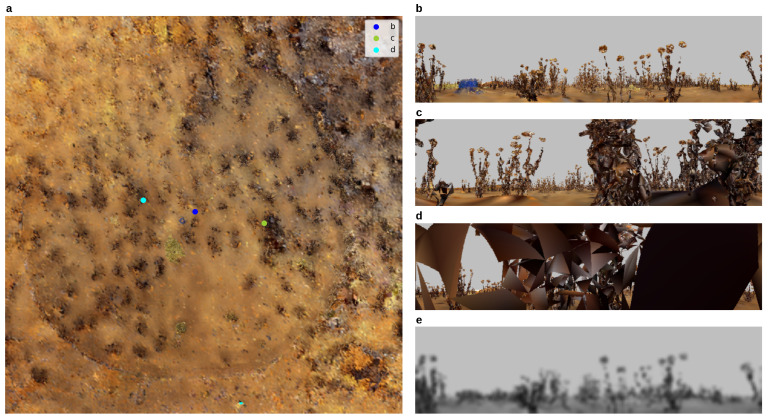
(**a**) Top-down view of the antworld 3D environment. It represents a 20 m×20 m desert ant’s environment model based on LIDAR scans. Sample views are shown from three different locations (coloured circles in (**a**); see legend). (**b**) A panoramic colour (RGB) image from the perspective of the agent from the dark blue circle. (**c**) An example of a partially occluded image (green circle). (**d**) An example of a fully occluded image where the agent is going through tussocks. (**e**) Image from (**b**) down-sampled to 2° per pixel (180 columns) and converted to grey-scale.

**Figure 2 biomimetics-09-00731-f002:**
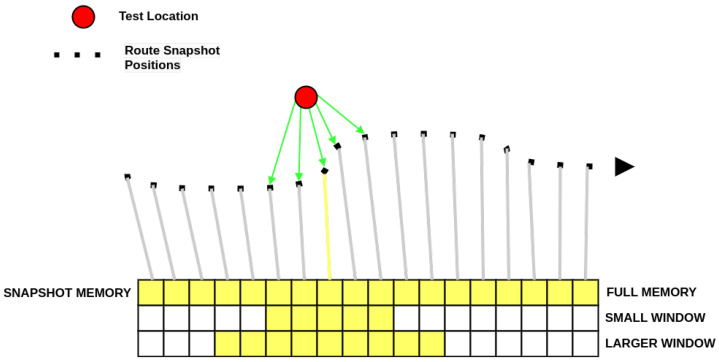
Schematic of the navigation algorithms. The training route (black squares) shows locations where snapshots are taken and stored in the snapshot memory. At a given test location, Perfect Memory (PM) compares the current view to all the training views, whereas SMW compares to a subset of the memory around the previous estimate of the location. This is indicated by small and larger window sizes highlighted in yellow in the 2nd and 3rd rows of the snapshot memory.

**Figure 3 biomimetics-09-00731-f003:**
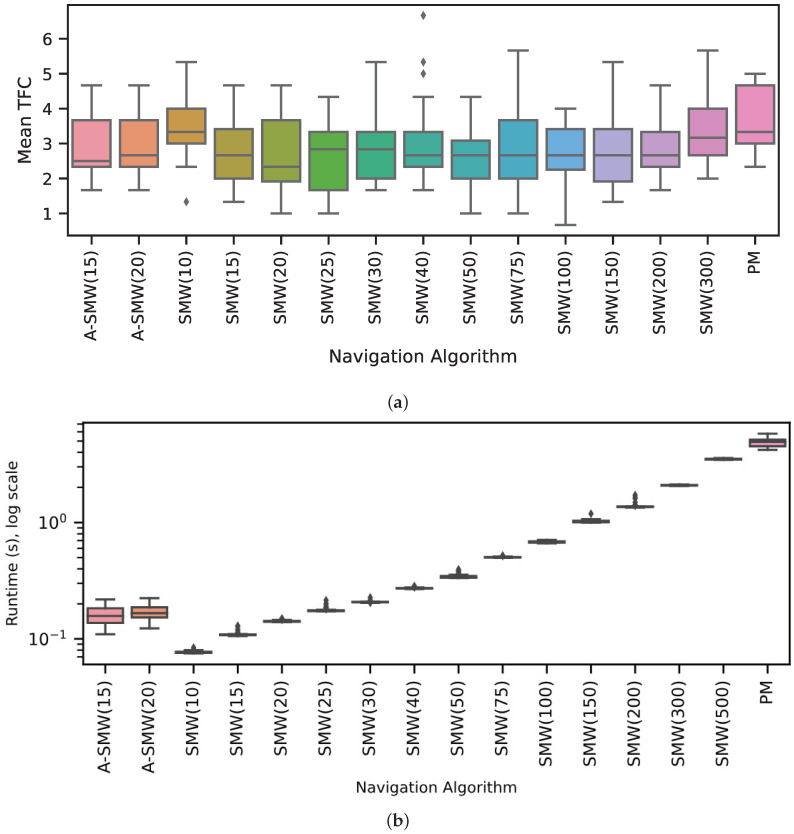
(**a**) Performance in terms of TFC for each algorithm across all routes. Boxplots show distributions of mean TFC over 20 routes where the mean is calculated over 3 trials for each route. Boxes show medians and quartiles, and whiskers show the range of most data apart from outliers (diamonds). Algorithms are shown by x-axis labels: SMW(10), SMW(15), etc., are fixed windows of length 10, 15, etc.; ASMW(15) and ASMW(20) are the adaptive methods with lengths initialised to 15 and 20, respectively; PM is the Perfect Memory algorithm (**b**) Runtime for all algorithms displayed as the distribution across 20 routes after averaging across repetitions but on a log-scale.

**Figure 4 biomimetics-09-00731-f004:**
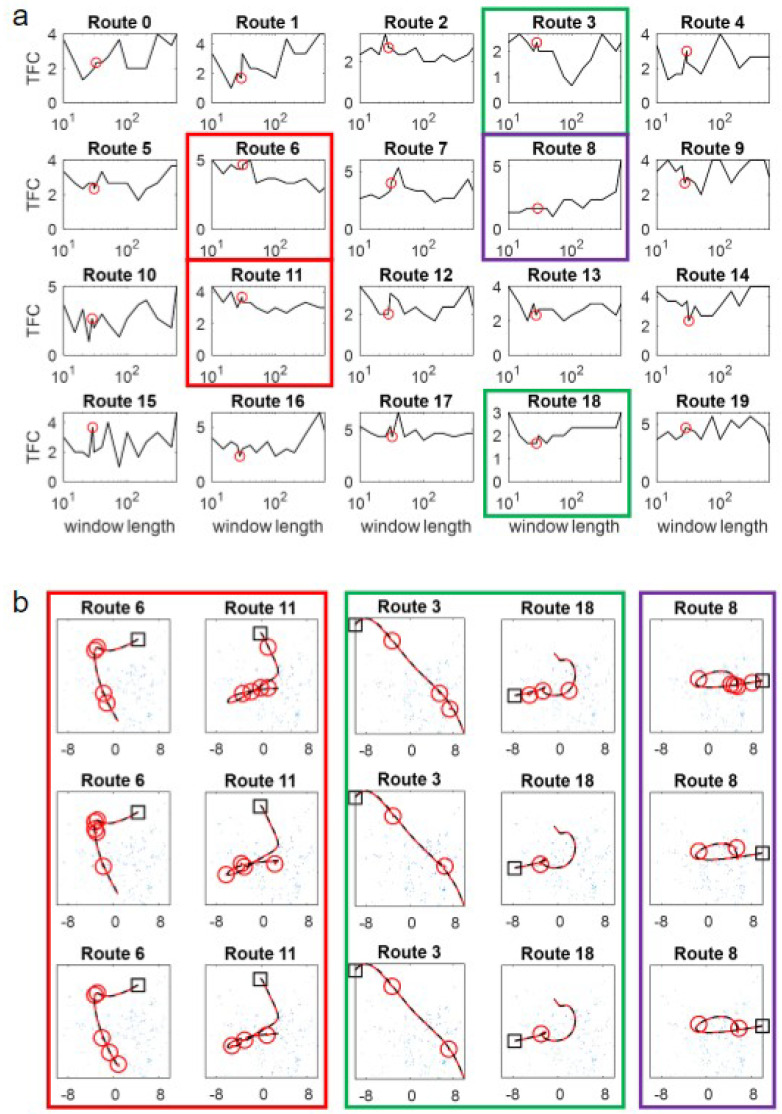
(**a**) Performance in terms of TFC for each route for each window. PM is shown as a window length of 600 (the number of training images) with window lengths shown on a log scale. ASMW(15) is shown as the red dot at the position on the x-axis, which is the average window length for that route. For clarity, ASMW(20) is not shown. As routes differ in difficulty, y-axes maxima are not set to be consistent. Coloured boxes highlight example routes shown in part (**b**). (**b**) Training (black dashed) and test (red) routes and positions of failures (red circles) for one repetition for PM (top row), SMW(20) (middle row) and ASMW(15) (bottom row). Black squares show start of routes. Tussock positions are indicated in blue. The example routes are chosen as they exhibit different trends in performance as a function of window size. Red boxes show routes that favour longer windows; green shows routes where an intermediate size is optimal; purple shows a route that is best for the shortest window.

**Figure 5 biomimetics-09-00731-f005:**
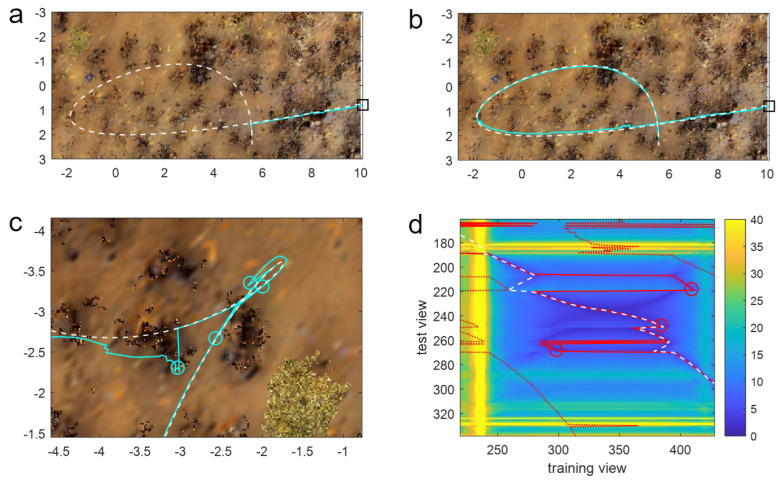
(**a**) The PM algorithm test path (cyan line; starting from black square) skips the main loop of the training path (white dashed line) at the loop intersection as it has no sequence information. This is a “valid” trajectory as the agent never left the vicinity of the route or failed to make progress. (**b**) SMW(50) avoids the intersection ambiguity issues and completes the route. Conventions as in (**a**). (**c**) Top-down view of a small looped part of route 16 and the trajectory of SMW(300). Conventions as in (**a**) but additionally cyan circles show the points of failure. (**d**) Heatmap showing the minimum IDF across all rotations of each test image (y-axis) with every training image (x-axis). Blue is a good match (low IDF); yellow is a bad match (high IDF, capped at 40 for visual clarity; see colour bar). Optimal match for each test view (defined as the closest training view) is a white-dashed line. Actual match plus window limits (150 either side of the match) are red lines (solid and dotted respectively), with failures as red circles.

**Figure 6 biomimetics-09-00731-f006:**
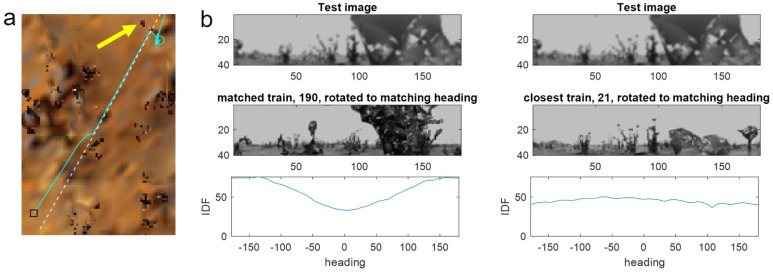
A partial occlusion causing aliasing failure for PM. (**a**) Top-down view of test (cyan) and training (white-dashed routes) showing the first failure (cyan o) and the tussock that causes the issue (yellow arrow). (**b**) **Top row**: test view at the point of diverting from the route. **Middle row**: best-matching training view (**left**) and closest training view (**right**), both rotated to the best matching heading. **Bottom row**: RIDF between test image and matched train (**left**) and between test and closest train (**right**). While both produce a good minimum, the left side is lower and hence picked by PM. However, the cause of the match is an ‘unlucky’ similar amount of occlusion seen from a random location further in the route at a random heading, leading the agent off-route and ultimately to failure. For the nearby position, there is more likelihood that the un-occluded parts of the image (i.e., the smaller tussocks on the left-hand side) drive the match leading to a correct heading.

**Figure 7 biomimetics-09-00731-f007:**
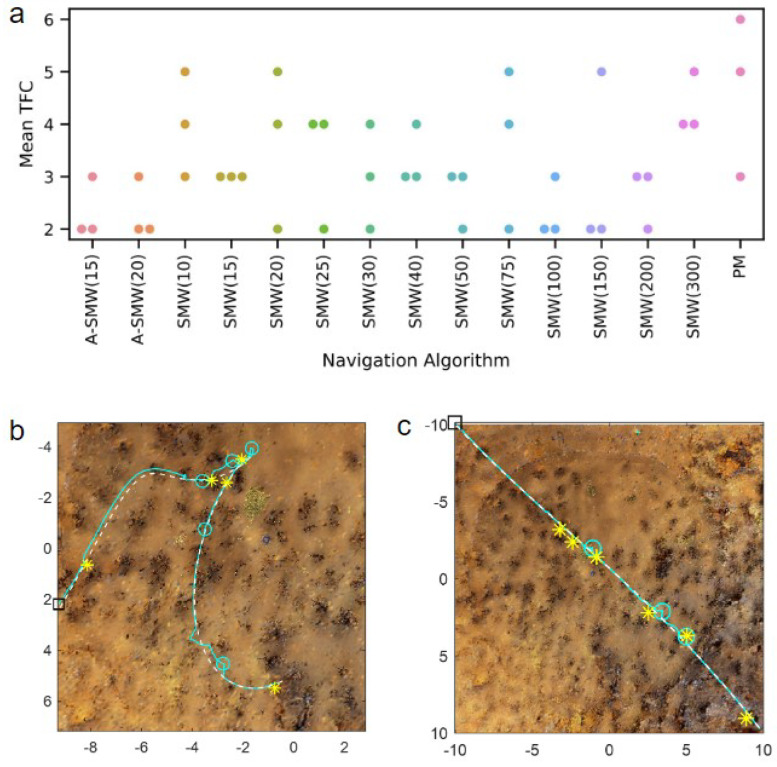
(**a**) Performance for all algorithms on route 16. Dots show TFC for each of three repetitions and are coloured according to the algorithm variant to match the colours in Figure 4. (**b**) Top-down view of one repetition of route 16 for SMW(10) additionally showing failures of PM from the same route and starting point. Training route; white-dashed line; Test route, cyan line; SMW(10) failures: cyan circles; PM failures, yellow stars. (**c**) Top-down view of one repetition for route 0 for SMW(10) additionally showing failures of PM. Conventions as in (**b**).

**Figure 8 biomimetics-09-00731-f008:**
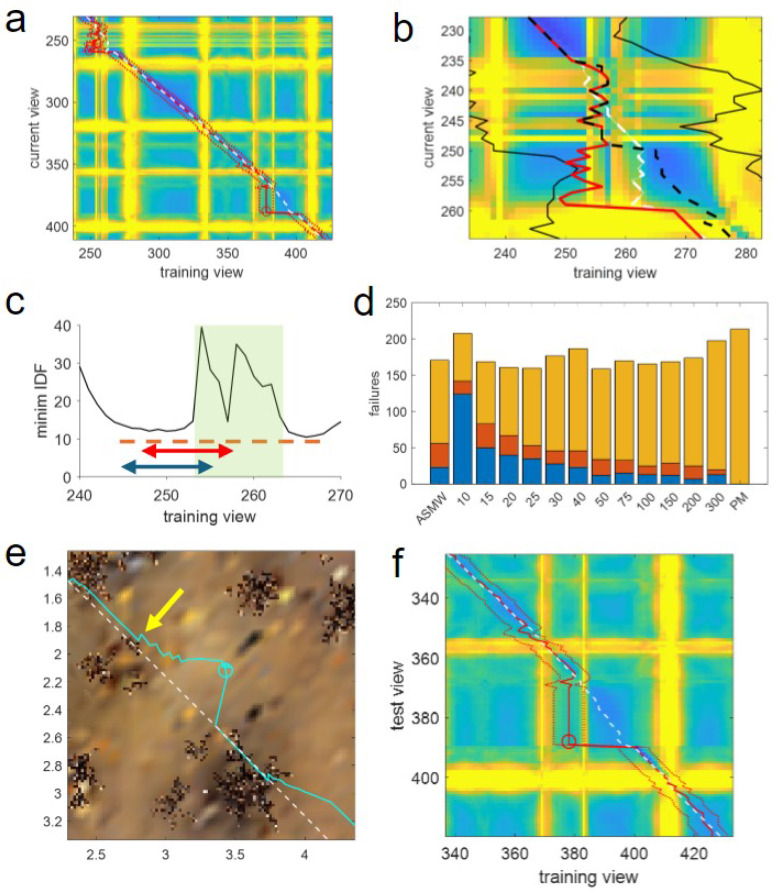
(**a**) Heatmap of route 0 showing current match and window limits for SMW(10) (red solid and dotted lines, respectively) and the optimal match (white dashed line). Heatmap colours as in Figure 5d. (**b**) Zoomed-in view of heatmap in (**a**) around the first failure additionally showing the ASMW(15) match and window limits (black dashed and solid lines respectively). SMW(10) window omitted for clarity. (**c**) Minimum difference between test image 248 (after exiting the grass) and nearby training images. Shaded green shows extent of the grass in the training images; red arrow shows the window of training views SMW(10) has available for matching on exiting the grass; blue arrow, shows the window for the next test image. Red dashed line highlights that the minimum after the grass is better quality (i.e., lower) than the one before. (**d**) The number of failures for different window lengths coloured according to type. Blue: failures when the window lags current agent position; Red: failures when window is ahead; Orange failures when optimal match is within the window. (**e**) Top-down view of the 2nd failure. Training route: white dashed line; Test path and failure: cyan line and circle; Yellow arrow: point where the agent gets a bad match due to minor occlusion and starts to deviate from optimal path (row 368 and column 378 in (**f**). (**f**) Heatmap around 2nd failure. Conventions and colours as in (**a**).

**Figure 9 biomimetics-09-00731-f009:**
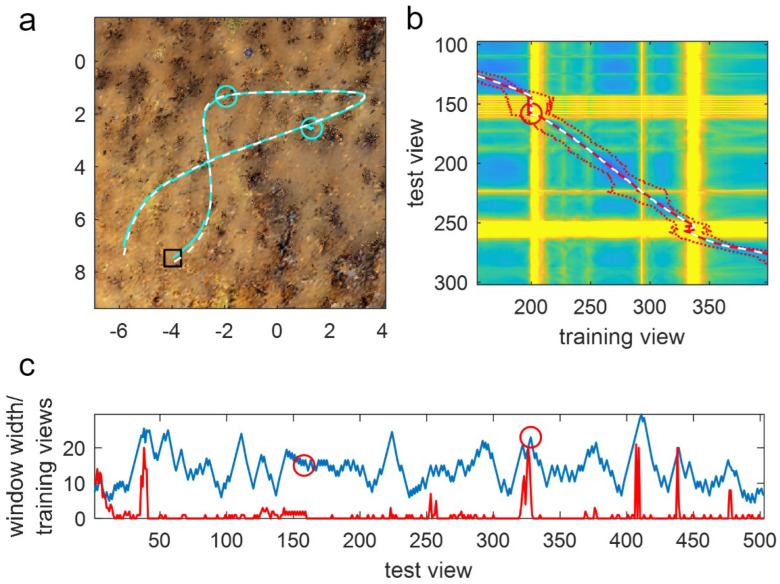
(**a**) Top-down view of ASMW(15) on route 8 showing training route (white-dashed line) and test route and failures (cyan line and circles), starting from black square. (**b**) Heatmap of ASMW(15) on route 8 shown in **a** around the first failure (red circle). Red solid and dotted lines show current match and window limits, respectively; White dashed line shows optimal match; Heatmap colours as in Figure 5d. (**c**) Window width (i.e., half the window length; blue line) and the difference in training view index between the matched view and the nearest view for each test view (red line). Failures are shown as red circles.

**Table 1 biomimetics-09-00731-t001:** Comparison of window-based methods used in this paper with other navigation algorithms. The sequence describes how or, indeed, whether, sequence information is used (see discussion for a more detailed description of the algorithms). The size is how many images are used in the sequences, and the image is the type of camera used while Nav. Type reflects whether the algorithm returns a direction (bearing) that the agent should move, or the agent’s location (VPR).

Algorithm	Sequence	Size	Image	Nav. Type
Multiple waypoints [29,30,31]	Implicit. Matches single snapshots from a set of waypoints	N/A	Panoramic	Bearing
View Sequence Route Representation [32,33]	Matches current to sequence around best match, only increments forwards	2	Forward Facing	Bearing
Play-back Navigation [34]	Matches current to sequence around the best match, only increments forwards	2	Panoramic	Bearing
SeqSLAM [28,35]	Matches current sequence to all possible training sequences	10–320	Forward Facing	VPR
SeqSLAM with Ensembles of Temporal Windows [36]	Matches current sequence to all possible training sequences	Fixed time (44–140 ms) worth of event frames	Forward Facing (event camera)	VPR
Multi-Process Fusion, SeqSLAM [37]	Matches current sequence to all possible training sequences. Image processing fusion.	5–20 or adaptive	Forward Facing	VPR
Visual Route Following aided by Odometry [38]	Matches ‘neighborhood’ of images near most likely robot location determined by the estimate of distance from start	Not specified but assume not many	Panoramic	Bearing
Bio-inspired Teach and Repeat Navigation [39]	Odometric estimate of location used to improve visual matching and reduce computation	6	Forward Facing	Bearing
Familiarity-based visual navigation [7]	No sequence	N/A	Panoramic	Bearing
**Current (SMW/ASMW)**	Matches current to sequence around the best match	10–300 or adaptive	Panoramic	Bearing

## Data Availability

Data is available at 10.25377/sussex.27890790.

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
