# Peer review of "Adaptive Route Memory Sequences for Insect-Inspired Visual Route Navigation"

_biomimetics, 2024, doi:10.3390/biomimetics9120731_

Round 1
Reviewer 1 Report
Comments and Suggestions for Authors
I think this paper has a rigorous structure and rich content, and the proposed path planning method is also very valuable. I have a few minor suggestions that I hope can be helpful for improving the paper:
1) In Figure 3, how is the amount of data used for each boxplot determined, and should it be increased to make the distribution more universally representative?
2) Additionally, the outlier markers in Figure 3(b) are a bit too large, making it unclear. It is suggested that a data table could be added for clarification.
3) It is also recommended to include more path planning results display similar to Figure 7.
Author Response
Comments 1: In Figure 3, how is the amount of data used for each boxplot determined, and should it be increased to make the distribution more universally representative?
Response 1: We apologise that this was only mentioned in figure label for figure 3 and very briefly. We have clarified how the values in the boxplot are calculated by adding to the methods and also say why we feel that 3 trials is representative p. 6, line 165-170
“Algorithm performance is assessed as the average TFC across three trials over 20 routes of increasing curvature, giving 20 mean TFC values. Each of the three trials starts at a different random starting position for each route. However, each algorithm variant is given the same starting position for each trial-route combination. We chose to use 3 trials as the reset behaviour of our TFC means that these results are representative of general behaviour.”
And also p. 5, line 127-128
“Each algorithm and window size is tested on 20 routes of increasing curvature with the performance averaged across three trials.”
We have also clarified the label for figure 3 on p.7:
“Performance in terms of TFC for each algorithm across all 20 routes. Boxplots show distributions of mean TFC over 20 routes where mean is calculated over 3 trials for each route”
Comments 2: Additionally, the outlier markers in Figure 3(b) are a bit too large, making it unclear. It is suggested that a data table could be added for clarification.
Response 2: We have reduced the size of the outliers on Figure 3 (a as well as b). We respectfully think that a data table is not needed for this figure after the clarifications
Comments 3: It is also recommended to include more path planning results display similar to Figure 7.
Response 3: We are a little unclear: all results figures apart from fig 3 have images showing paths of the agents. We could show more in fig 7 or make a new figure but not sure which of the routes are needed. We currently show examples from routes: 0, 3, 6, 8, 11, 16, 18. However we may be misunderstanding the reviewer
Reviewer 2 Report
Comments and Suggestions for Authors
This paper proposes an insect-inspired route navigation algorithm that uses stored panoramic images collected during a training route to subsequently derive directional information during route recapitulation. this paper tests the benefits of incorporating sequence information in our original algorithms. This work is very meaningful and highly innovative.
This paper is highly innovative, as it applies the concept of loop closure detection in robotics to path planning. I have extensive experience in visual loop closure or scene recognition, so I believe this paper offers sufficient novelty, and I understand its research methodology and ideas. Through experiments, the authors inform future researchers about which methods are feasible and what issues might arise, making this work highly valuable. Moreover, the authors have improved the experimental methods based on existing research.
There are some issues as follows.
1、What are the specific application scenarios for this method?
2、There are a few textual errors in the paper that should be carefully checked. For example, in line 66, is it 'irrelevant'?
Author Response
Comments 1:、What are the specific application scenarios for this method?
Response 1: Good point! Section added to the conclusions of the paper to outline potential use-cases and why this would be relevant p. 18, 536-543
“ As ASMW algorithms show performance benefits in terms of computation and are suitable for robots with only visual sensors, use-cases are focussed on GPS-denied or unreliable applications, as well as those where payload is limited such that Lidar sensors are not practical. In addition, it would be suited to environments in recognising location accurately (such as in dynamic our outdoor environments) is challenging. For instance, use-cases include space robotics, UAVs or small robots in unstructured outdoor environments in changing weather conditions, and indoor extreme environments such as nuclear inspections or decommissioning.”
Comments 2: There are a few textual errors in the paper that should be carefully checked. For example, in line 66, is it 'irrelevant'?
Response 2: Have fixed all we can find.
Reviewer 3 Report
Comments and Suggestions for Authors
The topic of navigation is very important for various engineering applications and will become more and more important. This manuscript contributes to the knowledge base. It does so from a very specific perspective: that of ants, which are very small and walk on the surface of the earth.
After a full introduction, the authors propose the technique to be used. Few assumptions are made and the whole process is fairly straightforward and easy to follow, even for a non-expert like me.
Various topics that follow logically from the initial results are addressed, and conclusions are drawn from the results.
My expertise is navigation in the maritime field and I would like to make two observations. In the analysis, the speed of the ant is included in the calculations. However, the images used are static. It seems logical to me to include the change in the optical array as a result of speed. This gives the heading and tells far from near. Far and near are both important for different aspects of navigation. And this last point is my second observation. In ship navigation we distinguish between control, tactical and strategic decisions. The manuscript does not explicitly make such an analysis. Is there a reason for this?
No comments
Author Response
Comments 1: My expertise is navigation in the maritime field and I would like to make two observations. In the analysis, the speed of the ant is included in the calculations. However, the images used are static. It seems logical to me to include the change in the optical array as a result of speed. This gives the heading and tells far from near. Far and near are both important for different aspects of navigation. And this last point is my second observation. In ship navigation we distinguish between control, tactical and strategic decisions. The manuscript does not explicitly make such an analysis. Is there a reason for this?
Response 1: We thank the reviewer for these two interesting points raised. The author is correct that optic flow information could be useful, perhaps for focussing attention on portions of the image. Indeed, we did trial an edge extraction algorithm in this environment, which would have a similar effect to basic optic flow extraction but the results were not vastly different than without edge extraction (which we felt was due to the fact we are working in simulation not on a real robot in this case. Hence we omitted this aspect so as to more clearly focus on the main aspect of the work, ie the differences seen from incorporating sequence. While the reviewer is correct that speed is implicit in the use of sequence in that the number and spacing of the images will vary according to agent speed and frequency of snapshots, as both parameters are fixed in our simulations, we do not discuss this analysis.
In terms of the maritime conventions, there could also be something interesting to do as we think is analogous to the moment to moment decision making that an animal might face while completing a higher level goal (eg obstacle avoidance while navigating and path planning etc.). In this case, as the reviewer notes, we are controlling only one aspect: navigation which is following the training route and are not trying to avoid shorter-scale obstacles, or optimise the overall path.
In both cases, as we feel that these issues would be most pertinent to a real-world robotic tes, we leave these aspects to future work which is discussed in in the conclusions in pp. 18-19: 544-556
"As these use-cases each have their own particularities and challenges, the intention of this work was to keep the implementation simple, so that benefits of the temporal window algorithms could be thoroughly investigated and applied to both ants and robots. However, this leaves various avenues for future work especially within robotic implementation, where more information could be incorporated depending on the robot's sensory capabilities. For instance, optic flow based visual processing could be used to highlight only the useful parts of the scene (typically those in the middle distance, not too near that they move quickly and not too far that they are essentially fixed). Additionally, idiothetic information about self-movement could be used to more accurately update the window position in combination with the best matched positions (similar to \cite{dall2021fast}). Further future avenues for robotic applications are to incorporate obstacle avoidance (to adjust small-scale details of the path) and path optimisation (to optimise the overall trajectory over multiple traversals) into the route following procedure."
Reviewer 4 Report
Comments and Suggestions for Authors
The authors have done a lot of work. A novel method is proposed and developed.
1) Contribution to the knowledge is unclear
1.1) Authors use a lot of "we"/"our" in the manuscript. This means that the algorithms and tests can only be done in this research group. A publication should show that knowledge can be used by anyone. So please use the third person only in the manuscript.
1.2) There is no section of conclusion
No conclusion means the work is not completed. Please add it on.
1.3) There is no future work
No future work means the author does not know the limitations of the method. Please add it on.
1.4) Many works are based on the author previous work, many reference are self reference. Authors need show the different of this work than previous work.
1.5) The author uses we/our in the review section, self-references, and references others. The author only provides a number, e.g., [12-13].
Therefore, it is difficult for reviewer to know their contributions and others contribution.
Please use the author's name in the review (Knaden developed an algorithm in......) 1) to show respect to the author who has done the previous work.
1.6) In the introduction, please use a table to compare the current method and show the method they use.
Comments on the Quality of English Language
The writing needs improved
Please do not use Oral English. "see", "Great", "Surprise"
e.g. The performance boost we see will be of no great surprise to the robotics community (line 447)
Author Response
Comments 1) Contribution to the knowledge is unclear
Response 1) We agree and thank the reviewer for the valuable suggestion of including a table which has allowed us to revise both introduction and discussion and clarify our contribution see (Response 1.6)
Comments 1.1) Authors use a lot of "we"/"our" in the manuscript. This means that the algorithms and tests can only be done in this research group. A publication should show that knowledge can be used by anyone. So please use the third person only in the manuscript.
Response 1.1) Changed throughout
Comments 1.2) There is no section of conclusion
No conclusion means the work is not completed. Please add it on.
Response 1.2) Apologies. Now added. P. 18-19, 530-556
“In summary, we have shown that incorporating sequence methods into familiarity-based algorithms can improve performance and reduce computation. Further, by assessing how sequence information is used, and comparing it to what we know about how temporal information is used by insects, the insights allow us to shed light on aspects of insect neuroethology and suggests improvements for robotic navigation via further bio-inspiration. As ASMW algorithms show performance benefits in terms of computation and are suitable for robots with only visual sensors, use-cases are focussed on GPS-denied or unreliable applications, as well as those where payload is limited such that Lidar sensors are not practical. In addition, it would be suited to environments in recognising location accurately (such as in dynamic our outdoor environments) is challenging. For instance, use-cases include space robotics, UAVs or small robots in unstructured outdoor environments in changing weather conditions, and indoor extreme environments such as nuclear inspections or decommissioning.
As these use-cases each have their own particularities and challenges, the intention of this work was to keep the implementation simple, so that benefits of the temporal window algorithms could be thoroughly investigated and applied to both ants and robots. However, this leaves various avenues for future work especially within robotic implementation, where more information could be incorporated depending on the robot's sensory capabilities. For instance, optic flow based visual processing could be used to highlight only the useful parts of the scene (typically those in the middle distance, not too near that they move quickly and not too far that they are essentially fixed). Additionally, idiothetic information about self-movement could be used to more accurately update the window position in combination with the best matched positions (similar to \cite{dall2021fast}). Further future avenues for robotic applications are to incorporate obstacle avoidance (to adjust small-scale details of the path) and path optimisation (to optimise the overall trajectory over multiple traversals) into the route following procedure.”
Comments 1.3) There is no future work
No future work means the author does not know the limitations of the method. Please add it on.
Response 1.3) Again apologies. Now added. pp. 19: 547-556
“However, this leaves various avenues for future work especially within robotic implementation, where more information could be incorporated depending on the robot's sensory capabilities. For instance, optic flow based visual processing could be used to highlight only the useful parts of the scene (typically those in the middle distance, not too near that they move quickly and not too far that they are essentially fixed). Additionally, idiothetic information about self-movement could be used to more accurately update the window position in combination with the best matched positions (similar to \cite{dall2021fast}). Further future avenues for robotic applications are to incorporate obstacle avoidance (to adjust small-scale details of the path) and path optimisation (to optimise the overall trajectory over multiple traversals) into the route following procedure.”
Comments 1.4) Many works are based on the author previous work, many reference are self reference. Authors need show the different of this work than previous work.
Response 1.4) We have reduced self citation counts. However, this is a relatively niche field and as a group we have done much of the work not only in modelling but also in the ant experiments hence it would be difficult to reduce these further.
Comments 1.5) The author uses we/our in the review section, self-references, and references others. The author only provides a number, e.g., [12-13].
Therefore, it is difficult for reviewer to know their contributions and others contribution.
Please use the author's name in the review (Knaden developed an algorithm in......) 1) to show respect to the author who has done the previous work.
Response 1.5) We apologise as we are more used to a non-numerical style of referencing. We have updated the review section comparing our work with others. P. 17, 469-472
“ The solution to this problem is to move from homing to individual waypoints and consider methods that set the bearing at which to move. Early work in this vein from Matsumoto et al. \cite{matsumoto1996visual}, Andersen et al. \cite{andersen1997appearance} and Tang and Yuta \cite{tang2001vision} show that navigation can be achieved with low computation by only attempting to match the next view with the current estimate plus one or two near neighbours, very neatly showing the benefits of short sequences.”
Comments 1.6) In the introduction, please use a table to compare the current method and show the method they use.
Response 1.6) We thank the reviewer for the valuable suggestion which serves to highlight the difference between our new method and others.
We have included a table in the introduction on p. 2. We discuss it on p. 3 line 65-69
“Aside from being applied to familiarity-based algorithms which have not previously had sequence incorporated, this work differs from navigation algorithms that have used sequence in two ways (Table 1). First a wide range of sequence lengths including an adaptive sequence length is tested and strengths and weaknesses analysed. Second, in contrast to many VPR algorithms, odometric information is not used in the SMW algorithm”
We have also extensively revised the text in the discussion to reflect better ow our work aligns within previous work P 17, 459-471:
“The improvement in performance is in line with other research demonstrating the benefits of incorporating either implicit or explicit information on sequence in navigation algorithms. For instance, several papers extended the range of single snapshot navigation by visually homing between a sequence of waypoints which implicitly utilises sequence information \cite{binding2006visual,vardy2006long,labrosse2007short,smith2007linked}. In this case, sequence information is included in the (typically small, i.e. around 10) discrete waypoints, but was not explicitly used in the matching process, hence sequence is referred to as being `implicit' for these methods in Table 1. While these methods can have impressive results \cite{labrosse2007short} there is evidence that the use of a strict sequence of images as attractors to a location is brittle \cite{smith2007linked}. The solution to this problem is to move from homing to individual waypoints and consider methods that set the bearing at which to move. Early work in this vein from Matsumoto et al. \cite{matsumoto1996visual}, Andersen et al. \cite{andersen1997appearance} and Tang and Yuta \cite{tang2001vision} show that navigation can be achieved with low computation by only attempting to match the next view with the current estimate plus one or two near neighbours, very neatly showing the benefits of short sequences. A third branch of work used long sequences of images to improve the robustness of visual place recognition algorithms, notably the SeqSLAM algorithm \cite{milford2012seqslam}. seqSLAM matches a sequence of current views to all possible sequences in the training views with the increased complexity of matching being compensated by increased performance for lower resolution images. ”
Comments 2); Comments on the Quality of English Language
The writing needs improved
Please do not use Oral English. "see", "Great", "Surprise"
e.g. The performance boost we see will be of no great surprise to the robotics community (line 447)
Response 2) We have changed this, p.17 line 459-461:
“The improvement in performance is in line with other research demonstrating the benefits of incorporating either implicit or explicit information on sequence in navigation algorithms. ”
However, in general we are happy with our style and have proofread the work by several native and non-native English speakers.
Round 2
Reviewer 4 Report
Comments and Suggestions for Authors
Authors provide references in their own work, e.g., Lines 73-81. This is not acceptable because it mixes their work with others' work.
Comments on the Quality of English LanguageSome oral English is used, and too-wording. And "We" and "Our" has been used too many times. Please remove all We and Our in the manuscript.
The proposed method in the paper should work without authors. Using "We and Our" is suggested that the method only works for the authors.
e.g. Line 445:
In this paper, we have explored and confirmed the computational and performance benefits of incorporating temporal information into familiarity-based navigation algorithms through route sequence information.
It can be replaced as "This paper explored ...."
Author Response
Comment 1: Ensure all references are relevant to the content of the manuscript.
Response 1: We have Reduced self-citations to the current level but all remaining references are needed. We cannot go further as this is a relatively niche field and as a group we have done much of the work not only in modelling but also in the ant experiments hence it would be difficult to reduce these further. All remaining references are needed.
Comment 2: Highlight any revisions to the manuscript, so editors and reviewers can
see any changes made.
Response 2: PDF with all changes highlighted has been uploaded
Comment 3: Provide a cover letter to respond to the reviewers’ comments and
explain, point by point, the details of the manuscript revisions.
Response 3: As well as responding individually, a pdf with all comments has been uploaded and is attached here
